# Insurance instability and use of emergency and office-based care after gaining coverage: An observational cohort study

**Paul R. Shafer**[1]*, **Stacie B. Dusetzina**[2], **Lindsay M. Sabik**[3], **Timothy F. Platts-Mills**[4], **Sally C. Stearns**[5], **Justin G. Trogdon**[5]

**1** Department of Health Law, Policy, and Management, School of Public Health, Boston University, Boston, Massachusetts, United States of America, **2** Department of Health Policy, School of Medicine, Vanderbilt University, Nashville, Tennessee, United States of America, **3** Department of Health Policy and Management, Graduate School of Public Health, University of Pittsburgh, Pittsburgh, Pennsylvania, United States of America, **4** Department of Emergency Medicine, School of Medicine, University of North Carolina at Chapel Hill, Chapel Hill, North Carolina, United States of America, **5** Department of Health Policy and Management, Gillings School of Global Public Health, University of North Carolina at Chapel Hill, Chapel Hill, North Carolina, United States of America

* pshafer@bu.edu

**Data Availability Statement:** This study is based on the publicly available Medical Expenditure Panel Survey Panel 18 Longitudinal Data File This study

## Abstract

### Background

The Affordable Care Act led to improvements in reporting a usual source of care, but it is unclear whether patients are changing their usual source of care in response to coverage gains. We assess whether prior insurance instability is associated with changes in use of emergency and office-based care after the Marketplace and Medicaid expansion were introduced.

### Methods

Our study draws from the 2013–14 Medical Expenditure Panel Survey, identifying a cohort of non-elderly adults with full-year health insurance coverage in 2014. We use linear and multinomial logistic regression to assess the relationship between insurance instability prior to 2014 (uninsured for 1–11 months, ≥12 months) and person-level changes in use of health care after gaining coverage (change in ED and office visits from 2013 to 2014) with continuously insured individuals serving as a comparison group.

### Results

Being uninsured for at least one year prior to gaining full-year coverage in 2014 was associated with a 33% increase in ED visits (0.06 visits, p<0.01) and a 47% increase in office visits (1.10 visits, p<0.01), driven by those gaining public coverage. We found no evidence of substitution across settings in the short term, often a stated goal of expansion.

is based on the publicly available Medical Expenditure Panel (MEPS) Survey Panel 18 Longitudinal Data File (https://meps.ahrq.gov/mepsweb/data_stats/download_data_files_detail.jsp?cboPufNumber=HC-172) and 2012 National Health Interview Survey (NHIS) Sample Adult File (https://www.cdc.gov/nchs/nhis/nhis_2012_data_release.htm). The linkage of respondent identifiers between these two data sets as well as the state and county identifiers used to merge on county-level characteristics are only available upon approval of a restricted data use proposal to the Agency for Healthcare Research and Quality (AHRQ) for analysis in a Federal Statistical Research Data Center (RDC). AHRQ provides a description about the confidential and non-public use variables available in the RDC (https://meps.ahrq.gov/mepsweb/data_stats/onsite_datacenter.jsp) and documentation for the NHIS-MEPS linkage (https://meps.ahrq.gov/mepsweb/data_stats/download_data_files_detail.jsp?cboPufNumber=NHIS%20Link) on their web site. Proposals for access to restricted variables (https://meps.ahrq.gov/data_stats/data_center_application.pdf) should be submitted via email to cfactdc@ahrq.hhs.gov.

**Funding:** This study was supported by grants from the Robert Wood Johnson Foundation (73923, https://www.rwjf.org) and Horowitz Foundation for Social Policy (https://www.horowitz-foundation.org) to PRS. The funders had no role in study design, data collection and analysis, decision to publish, or preparation of the manuscript.

**Competing interests:** The authors have declared that no competing interests exist.

## Conclusion

The long-term uninsured may have substantial health needs and pent-up demand for health care, seeing more physicians across multiple settings in the year after gaining coverage as they seek to get unmanaged conditions under control. Closing the gap in primary care use between the previously uninsured and those with health insurance coverage may help improve long-term health outcomes.

## Introduction

The Patient Protection and Affordable Care Act of 2010 (ACA) took a multi-pronged approach to expansion of health insurance coverage (dependent coverage up to age 26, Medicaid expansion, and subsidized coverage in the Marketplace) that has covered approximately 20 million Americans [1]. The first few years after the introduction of the Marketplace and Medicaid expansion in 2014 showed substantial declines in uninsurance, evidence of improvements in self-reported health and preventive service use, gains in reporting a usual source of care, and other access to care measures [2–5]. However, it is unclear though whether patients actually are changing where they use care in response to gaining health insurance.

Before the ACA, the majority of the evidence describing the effects of coverage on use of care was based on expanded eligibility for public coverage with the exception of the 2006 Massachusetts health reform, the model for the ACA. These studies yielded mixed findings about whether coverage expansions alone encourage less use of emergency departments (ED) and greater use of primary care [6–14]. More recent evidence has shown that the state and local contexts for insurance expansion matter a lot (e.g., baseline rates of coverage, health of the target population), accounting for these seemingly contradictory findings in the past [15]. Longer periods of being uninsured have been associated with lower access to care and those with unstable coverage are more likely to use the ED as a primary source of care, refill prescriptions less frequently, and have uncontrolled hypertension [16–20]. Disparities in insurance instability by race and ethnicity help compound difficulties in access to care and help maintain health inequities, though there is evidence of reduced Hispanic insurance instability post-ACA [20, 21]. However, the relationship between how long someone went uninsured before gaining coverage, what we refer to as insurance instability, and subsequent changes in health care use after gaining coverage has not yet been studied post-ACA.

The severity of insurance instability in a population targeted for coverage expansion plays a role in how much and where they use care, which is important for their cost of care and future health outcomes. Extensive research has examined the effects of coverage expansion on specific settings of care (e.g., inpatient stays, emergency department visits, office visits) [6–14], often comparing population-average changes in the use of different settings to draw conclusions about whether or not patients are substituting between them. This can lead to attribution errors because changes are not being observed within individuals across settings and time. We build upon this by measuring substitution from ED to office-based settings using person-level changes in health care use within a nationally representative cohort of adults that coincides with widespread changes in access to health insurance coverage.

## Methods

### Data

This study uses a restricted linkage between the 2012 National Health Interview Survey (NHIS) and the 2013 and 2014 Medical Expenditure Panel Survey (MEPS) as its primary data

source [22, 23]. These years were chosen due to the implementation of the two largest ACA coverage expansions in 2014 (Medicaid expansion and Marketplace), providing a large number of individuals transitioning from uninsurance into coverage. We also use data from the Area Health Resources File (AHRF) to capture county-level demographic characteristics and the availability of health care facilities and providers [24]. We use data from the Centers for Medicare and Medicaid Services' Marketplace Public Use Files and the Kaiser Family Foundation to create measures of Marketplace plan availability and costs in 2014, the first year in which the health insurance exchanges were in operation [25–27]. This study was approved as exempt by the Non-Biomedical Institutional Review Board at the University of North Carolina at Chapel Hill (#17–0774).

## Sample

Fig 1 shows the inclusion criteria used in our analysis and how it affected our final sample size. We focused on nonelderly adults (18 to 64 years of age in both years) to isolate those individuals whose health insurance coverage status and health care use would be most affected by the implementation of the Marketplace and Medicaid expansion. We dropped individuals who were not present in all 5 rounds of data collection due to death, military service, or

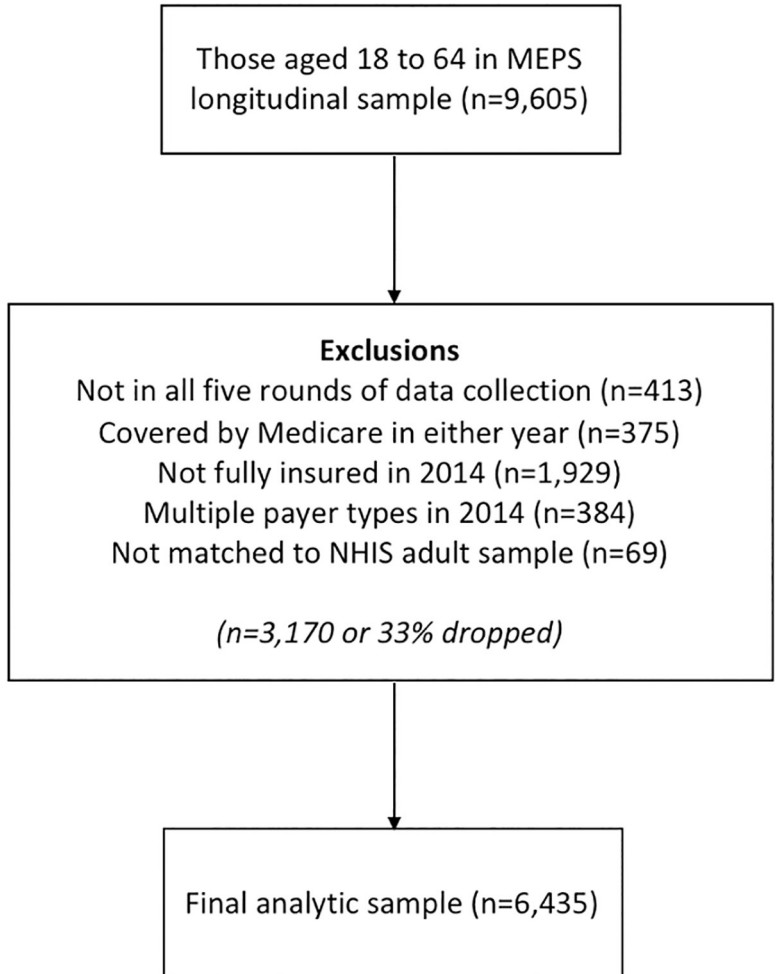

**Fig 1. Sample size and inclusion criteria.**

incarceration. Those attriting from the MEPS sample for other reasons are already excluded with appropriate adjustment to the survey weights to maintain representativeness [28]. We also dropped anyone covered by Medicare in either year (e.g., dual-eligible, disabled). We only included those who had full-year coverage with a single insurance type (i.e., private, public) during 2014, regardless of their prior insurance instability. We then matched this MEPS analytic sample to the 2012 NHIS adult sample, yielding an analytic sample of 6,435 individuals or approximately 67% of the original age-restricted sample.

## Measures

MEPS provides annual visit volumes for various health care settings for each person in the sample, regardless of their insurance status, allowing us to quantify changes in use even for those who were previously uninsured. To assess changes in the volume of use of ED and office-based care, we first calculated the year-over-year change (number of visits in 2014 minus the number of visits in 2013) for each setting separately for each person in our sample. We defined office-based care in each year as the total number of office visits, as opposed to outpatient clinic visits, with a physician, physician assistant, nurse practitioner, or nurse. This captures not only primary care in a family medicine clinic or community health center, but is flexible enough to include management of chronic conditions at a multispeciality clinic. Then, we generated a four-level categorical outcome to measure potential year-over-year substitution (from 2013 to 2014) between settings with the following mutually exclusive categories: 1) increase in both ED and office visits; 2) same or fewer ED visits with an increase in office-based visits; 3) increase in ED visits with the same or fewer office visits; and 4) same or fewer of both ED and office visits. We chose this categorical measure over a proportion of total visits-based approach since it does not result in patients with zero visits dropping out of the model.

Insurance instability prior to 2014 was the explanatory variable of interest, defined as being continuously insured (insured for all of 2013); short-term uninsured (uninsured for at least one month in 2013 but not the full year); or long-term uninsured (uninsured for all of 2013 or longer) [29, 30]. Among those categorized as long-term uninsured, over 80% (81.7%) were continuously uninsured for at least 36 months prior to January 2014, providing a strong justification for considering this group as meaningfully different in health insurance experience than those only uninsured for a short time. We used several individual and family-level demographic characteristics in our analysis, including age, sex, race/ethnicity, education, employment status, marital status, household size, and family income as a percentage of the federal poverty level. We created an indicator of self-reported health decline based on the year-over-year change (2013 to 2014) in perceived health status and captured insurance type in 2014 (i.e., private, public) to look for differential effects by benefit design (e.g., exposure to significant cost-sharing).

We included several state and county-level factors associated with access to care, health insurance, and the socioeconomic environment in our analysis. Quartiles for the number of hospitals with an ED, number of primary care physicians, number of physician extenders, number of federally qualified health centers, and whether the county was a health professional shortage area were used to describe the availability of providers in each county [31]. We also included unemployment rate, percentage of population non-white, in poverty, and uninsured, and whether the county was non-metro to account for the socioeconomic environment.

## Statistical analysis

We use a partially differenced estimation approach, with outcomes that are year-over-year changes in visit volume or our categorical measure that captures year-over-year substitution between settings, along with time-invariant controls. For the volume outcomes, we used

separate linear regression models to estimate the relationship between insurance instability and changes in ED and office visits after gaining coverage, controlling for individual-, family-, and area-level characteristics. We estimated the difference in the year-over-year change in ED and office visits for the short- and long-term uninsured relative to the continuously insured group. Similarly, we used multinomial logistic regression models to estimate the association between insurance instability and substitution across settings. We estimated the percentage point change (average marginal effects) in the predicted probability of being in each of the four mutually exclusive year-over-year substitution categories for the short- and long-term uninsured relative to the continuously insured group. We include state fixed effects to account for any remaining time-invariant unobservable differences between states and used the survey weights provided to account for the complex sampling design. We ran stratified versions of these analyses by insurance type gained in 2014 (public, private), adding additional state-level factors to the model including Medicaid expansion status, number of Marketplace insurers, and the average benchmark premium to account for differences in plan competition and pricing across states. We also performed numerous sensitivity analyses, including 1) raising the lower bound on age to 27 (as the dependent coverage provision allowed parents to include children up to the age of 26), 2) dropping those with family incomes over 400% of the federal poverty level (not eligible for advanced premium tax credits in the Marketplace), and 3) excluding those with no visits in both years (used at least one ED or office visit in each year). All analyses were conducted in Stata 15 for Linux [32].

## Results

The two uninsured groups were significantly younger on average than the continuously insured group (Table 1). Sex was fairly even split except for the short-term uninsured group, which skewed more heavily female. The percentage of each group that is white non-Hispanic decreases as insurance instability increased. Educational attainment, being employed, being married, and household income were all negatively associated with insurance instability. We do not find significant differences in county-level number of hospitals with an ED, number of primary care physicians, or number of physician extenders by insurance instability. We do observe lower availability of federally qualified health centers within the short- and long-term uninsured groups. Those in the two uninsured groups were also significantly more likely to live in a county with a higher percentage of its population non-white, unemployed, in poverty, and uninsured. We find no significant differences between groups by health professional shortage area status or being in a non-metro area.

ED use is generally low with an average of approximately 0.2 visits per person per year, increasing slightly with insurance instability (S1 Fig). In the baseline year (2013), the continuously insured group averaged approximately one additional office visit per year (3.3) than the short-term (2.2) and long-term uninsured (2.3) groups. Fig 2 shows survey weighted regression adjusted year-over-year change in ED and office visits by insurance instability. We find no year-over-year change in ED use for the continuously insured and short-term uninsured groups, but observe a small increase in ED use for the long-term uninsured group (0.06 visits per person, p<0.01). We observe small changes in year-over-year office visits for the continuously insured (0.10 visit increase, p<0.01) and short-term uninsured (0.23 visit decrease, p = 0.02) groups, but more than a one visit per year increase for the long-term uninsured group (1.10 visits per person, p<0.01). This sizable increase for the long-term uninsured group closes the gap with the continuously insured group once covered in 2014. These estimates are consistent with those from our linear regression models of year-over-year change in ED and office visits (S1 and S2 Tables).

**Table 1. Weighted sample characteristics by insurance instability prior to 2014, United States, 2013–2014.**

| Characteristic | % or mean (standard error) | | | |
|---|---|---|---|---|
| | Overall | By insurance instability prior to 2014 | | |
| | | Continuously insured | Short-term uninsured (1–11 months) | Long-term uninsured (≥12 months) |
| Number of observations | 6,435 | 5,555 | 251 | 629 |
| *Individual-level* | | | | |
| Age* | 41.2 (0.2) | 41.6 (0.3) | 36.0 (1.1) | 38.9 (0.7) |
| Sex* | | | | |
|   Female | 52.5% | 52.4% | 61.2% | 49.9% |
|   Male | 47.5% | 47.6% | 38.8% | 50.1% |
| Race/ethnicity* | | | | |
|   White, non-Hispanic | 65.7% | 67.3% | 54.7% | 50.9% |
|   African American, non-Hispanic | 11.4% | 10.9% | 19.2% | 13.9% |
|   Hispanic | 13.7% | 12.5% | 18.5% | 26.8% |
|   Other, non-Hispanic | 9.1% | 9.3% | 7.6% | 8.3% |
| Education* | | | | |
|   High school or less | 31.0% | 29.1% | 40.9% | 49.1% |
|   Some college or more | 69.0% | 70.9% | 59.1% | 50.9% |
| Employed* | 79.2% | 80.6% | 68.7% | 66.4% |
| Married* | 57.6% | 60.3% | 35.8% | 34.1% |
| Household size | 3.0 (0.04) | 3.0 (0.04) | 2.8 (0.2) | 2.9 (0.1) |
| Household income relative to the federal poverty level* | | | | |
|   0–99% | 9.4% | 7.7% | 20.5% | 24.5% |
|   100–199% | 13.7% | 11.5% | 23.0% | 36.0% |
|   200–400% | 30.1% | 30.1% | 33.8% | 28.0% |
|   >400% | 46.8% | 50.6% | 22.7% | 11.6% |
| Perceived decline in health *(from 2013 to 2014)** | 22.8% | 22.0% | 27.3% | 30.2% |
| Had an ambulatory care sensitive condition *(in 2013)** | 30.4% | 31.1% | 23.5% | 24.9% |
| *County-level* | | | | |
| Number of hospitals with an ED *(quartiles)* | | | | |
|   1 –top | 41.5% | 41.7% | 37.1% | 41.3% |
|   2 | 17.6% | 18.0% | 16.4% | 13.6% |
|   3 | 22.6% | 22.8% | 23.3% | 20.4% |
|   4 –bottom | 18.3% | 17.5% | 23.1% | 24.7% |
| Number of primary care physicians *(quartiles)* | | | | |
|   1 –top | 30.9% | 30.9% | 29.9% | 31.6% |
|   2 | 29.2% | 29.5% | 26.9% | 26.0% |
|   3 | 21.6% | 21.8% | 19.4% | 20.0% |
|   4 –bottom | 18.4% | 17.9% | 23.9% | 22.4% |
| Number of physician extenders *(quartiles)* | | | | |
|   1 –top | 31.0% | 31.0% | 28.3% | 31.6% |
|   2 | 29.0% | 29.3% | 25.0% | 26.7% |
|   3 | 20.5% | 20.4% | 25.0% | 19.5% |
|   4 –bottom | 19.6% | 19.3% | 21.8% | 22.2% |
| Number of federally qualified health centers *(quartiles)** | | | | |

*(Continued)*

**Table 1.** (Continued)

| Characteristic | % or mean (standard error) | | | |
|---|---|---|---|---|
| | Overall | By insurance instability prior to 2014 | | |
| | | Continuously insured | Short-term uninsured (1–11 months) | Long-term uninsured (≥12 months) |
| 1 –top | 34.8% | 35.3% | 29.8% | 31.4% |
| 2 | 26.2% | 26.7% | 27.2% | 20.1% |
| 3 | 23.4% | 22.9% | 22.0% | 29.9% |
| 4 –bottom | 15.6% | 15.2% | 21.0% | 18.7% |
| Percentage of county population non-white* | 34.3% | 33.9% | 38.0% | 36.8% |
| Percentage of county population unemployed* | 7.3% | 7.2% | 7.8% | 7.7% |
| Percentage of county population in poverty* | 15.5% | 15.3% | 17.1% | 16.8% |
| Percentage of county population uninsured* | 19.5% | 19.2% | 22.3% | 21.9% |
| Health professional shortage area | 35.8% | 35.3% | 38.5% | 40.0% |
| Non-metro area | 13.9% | 13.8% | 13.0% | 16.2% |

* p<0.05 for differences across groups

Table 2 shows the predicted probabilities and percentage point changes in predicted probability for each substitution category by insurance instability prior to 2014. Nearly 60% of people across all three groups (continuously insured, short-term uninsured, long-term uninsured) are predicted to stay the same or decrease use in both settings with very little variation between groups. Prior insurance instability was not associated with substitution across settings. Regression results for the substitution outcome are shown in S3 Table and the sensitivity analyses described above are shown in S4 to S6 Tables.

Table 3 shows the association of insurance instability with each of the three outcomes stratified by type of insurance gained (public or private). We find that being on public insurance appears to be driving the small increase in ED use and large increase in office visits that we observed generally among the long-term uninsured. For the long-term uninsured who transitioned to public insurance (e.g., Medicaid) in 2014, we observe a 0.24 visit increase in ED use (p<0.01) and a 1.94 visit increase in use of office-based care (p = 0.08) year-over-year. We find

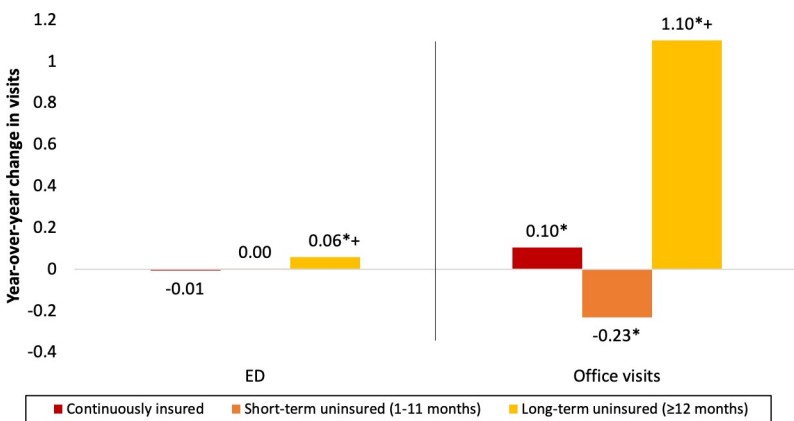

**Fig 2. Weighted adjusted year-over-year change in emergency department and office visits by insurance instability, United States, 2013–2014.** N = 6,371. * p<0.05 for year-over-year change not equal to zero. + p<0.05 for difference from continuously insured group.

**Table 2. Year-over-year substitution by insurance instability prior to 2014, United States, 2013–2014.**

| Change in visits from 2013 to 2014 | Predicted probabilities | | | Percentage point change (95% confidence interval) | |
|---|---|---|---|---|---|
| | Continuously insured | Short-term uninsured (1–11 months) | Long-term uninsured (≥12 months) | Short-term uninsured (1–11 months) | Long-term uninsured (≥12 months) |
| > ED visits, > office visits | 5.0% | 4.3% | 6.6% | -0.4 (-3.8, 3.0) | 1.2 (-1.1, 3.4) |
| ≤ ED visits, > office visits | 31.9% | 28.7% | 29.1% | -1.8 (-9.4, 5.8) | 0.3 (-5.0, 5.7) |
| > ED visits, ≤ office visits | 3.7% | 8.1% | 5.5% | 3.6 (-1.0, 8.2) | 0.5 (-1.9, 2.8) |
| ≤ ED visits, ≤ office visits | 59.4% | 58.9% | 58.7% | -1.4 (-9.1, 6.4) | -2.0 (-7.7, 3.7) |

For predicted probabilities, each column sums to 100%. For percentage point changes, each column sums to 0. Percentage point changes represent the average marginal effect estimates from multinomial logistic regression and are relative to the continuously insured group. These models include individual and county-level characteristics shown in Table 1 as covariates, the coefficient estimates on which these results are based are shown in Appendix Table 3. N = 6,371.

no changes among those who were long-term uninsured and gained private insurance, or those who were short-term uninsured regardless of insurance type gained in 2014. For the substitution outcome, we also observe a large negative association (–8.9 percentage points) of long-term uninsurance and gaining public insurance with having the same or fewer of both ED and office visits year-over-year, which is indicative of a shift towards increased use of one or both settings and consistent with the results of our setting-specific models.

## Discussion

Our study focuses on whether insurance instability, or the length of time that a person was continuously uninsured, influences changes in health care use once they gain coverage. We find that those who had been uninsured for a year or longer increased both the amount of ED and office visits used after gaining health insurance in 2014. These changes were driven by those gaining public insurance (e.g., Medicaid) with no significant changes among those gaining private insurance, perhaps due to high deductibles and other cost-sharing not present under public coverage. The large relative increase in ED visits after gaining coverage for those who had been uninsured for more than a year generally matches the findings of the Oregon Health Insurance Experiment in which those gaining Medicaid increased their ED use [14, 33]. A strength of our study is the use of restricted data that allows us to account for county-level health care facility and provider availability for each person in the sample, allowing us to avoid overestimating increases in ED use attributable to poor primary care access.

This analysis has several limitations. Given the observational nature of this study, we are unable to make any causal claims about the relationships observed nor can we qualitatively describe how newly insured patients are making decisions about their use of care. For example, what are their barriers to care aside from coverage? Only having one year on each side of the coverage transition is another limitation due to potential changes in health or regression to the mean, we account for this by controlling for changes in perceived health between years. Perceived health allows us to capture changes in health that using do not rise to the level of a serious diagnosis (e.g., cancer, cardiovascular disease, diabetes) but still may result in changes in health care use. This is also a short timeframe over which to observe change, a patient may need long-term access to primary and specialty care before substantial shifts away from ED use would potentially occur. Similarly, we do not know if the changes that we observe will

**Table 3. Stratified estimates of insurance instability and year-over-year changes in use of emergency departments and office visits, United States, 2013–2014.**

| Outcome | Percentage point change (95% confidence interval) | |
|---|---|---|
| | Insurance type (2014) | |
| | Private | Public |
| *ED visits (change in visits)* | | |
| Continuously insured | – | – |
| Short-term uninsured | 0.07 (-0.02, 0.16) | 0.09 (-0.13, 0.30) |
| Long-term uninsured | 0.03 (-0.03, 0.09) | 0.24** (0.07, 0.41) |
| *Office visits (change in visits)* | | |
| Continuously insured | – | – |
| Short-term uninsured | -0.31 (-1.00, 0.38) | -0.44 (-2.42, 1.55) |
| Long-term uninsured | 0.06 (-0.34, 0.47) | 1.94 (-0.26, 4.14) |
| *Substitution (percentage point change in predicted probability)* | | |
| > ED visits, > office visits | | |
| Continuously insured | – | – |
| Short-term uninsured | -0.5 (-4.7, 3.7) | 1.0 (-8.3, 10.2) |
| Long-term uninsured | -0.7 (-3.1, 1.7) | 2.9 (-1.9, 7.7) |
| ≤ ED visits, > office visits | | |
| Continuously insured | – | – |
| Short-term uninsured | -3.3 (-11.3, 4.8) | 0.6 (-12.8, 14.0) |
| Long-term uninsured | -1.6 (-8.0, 4.8) | 2.4 (-6.1, 11.0) |
| > ED visits, ≤ office visits | | |
| Continuously insured | – | – |
| Short-term uninsured | 2.8 (-1.6, 7.3) | 4.6 (-2.5, 11.8) |
| Long-term uninsured | -0.8 (-3.2, 1.5) | 3.6 (-1.0, 8.2) |
| ≤ ED visits, ≤ office visits | | |
| Continuously insured | – | – |
| Short-term uninsured | 1.0 (-7.8, 9.7) | -6.2 (-20.1, 7.6) |
| Long-term uninsured | 3.1 (-4.0, 10.2) | -8.9* (-17.6, -0.2) |

* p<0.05

** p<0.01

The model for those with private insurance also includes the number of Marketplace insurers in the state, and state average benchmark premium in 2014. The model for those with public insurance also includes state Medicaid expansion status. These controls are in addition to the individual- and county-level characteristics shown in Table 1. N = 6,371.

persist for future years. For example, previous work has shown that large increases in new patient primary care visits among those switching payers and plans diminish over time [34].

We find compelling evidence that expanding coverage to the long-term uninsured yields significant increases in health care use, but not necessarily the desired substitution from the ED to office-based settings. As federal and state policymakers consider changes to cover some or all of the remaining uninsured (e.g., smoothing of the Marketplace subsidy cliff, Medicaid expansion, Medicaid buy-in, public option), insurance instability may be an important piece of the puzzle in projecting the health care use of those newly covered. Prior expansions, such as Massachusetts health reform, have increased the insured rate without increasing the frequency of plan switches [35], another dimension of instability that can be related to access barriers, and reducing coverage gaps for some populations [36]. Closing the gap in primary care

use may help improve long-term health of those previously uninsured. The long-term uninsured may have substantial health needs and pent-up demand for health care, seeing more physicians across multiple settings after gaining coverage as they seek to get unmanaged conditions under control. The large increase in office visits that we observe for this group puts their use of office-based care on par with the continuously insured, an important first step to closing disparities in prevention and early detection of disease between the insured and uninsured. Inconsistent findings for the short-term uninsured group warrant further study given their implications for continuity of care, particularly as more states begin to experiment with Medicaid work requirements that will likely increase transitions in and out of coverage.

## Supporting information

**S1 Fig. Weighted average emergency department and office visits by insurance instability prior to 2014, United States, 2013–2014.** N = 6,371.
(TIF)

**S1 Table. Weighted linear regression estimates of year-over-year change in emergency department visits, United States, 2013–2014.**
(DOCX)

**S2 Table. Weighted linear regression estimates of year-over-year change in office visits, United States, 2013–2014.**
(DOCX)

**S3 Table. Weighted multinomial logistic regression estimates of year-over-year substitution, United States, 2013–2014.**
(DOCX)

**S4 Table. Sensitivity analysis of year-over-year change in emergency department visits, United States, 2013–2014.**
(DOCX)

**S5 Table. Sensitivity analysis of year-over-year change in office visits, United States, 2013–2014.**
(DOCX)

**S6 Table. Sensitivity analysis of year-over-year substitution, United States, 2013–2014.**
(DOCX)

## Acknowledgments

Access to the NHIS-MEPS linkage and restricted MEPS variables was provided by the Center for Financing, Access, and Cost Trends at the Agency for Healthcare Research and Quality for analysis at the Triangle Census Research Data Center at Duke University in Durham, North Carolina. The results and conclusions in this paper are those of the authors and do not indicate concurrence by the Census Bureau, Agency for Healthcare Research and Quality, or the Department of Health and Human Services.

## Author Contributions

**Conceptualization:** Paul R. Shafer, Stacie B. Dusetzina, Lindsay M. Sabik, Timothy F. Platts-Mills, Sally C. Stearns, Justin G. Trogdon.

**Formal analysis:** Paul R. Shafer.

**Funding acquisition:** Paul R. Shafer.

**Writing – original draft:** Paul R. Shafer.

**Writing – review & editing:** Paul R. Shafer, Stacie B. Dusetzina, Lindsay M. Sabik, Timothy F. Platts-Mills, Sally C. Stearns, Justin G. Trogdon.

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
