## [Decision Letter · Decision Letter 0]

18 Jun 2020

PONE-D-20-09519

Insurance instability and use of emergency and office-based care after gaining coverage: an observational cohort study

PLOS ONE

Dear Dr. Shafer,

Thank you for submitting your manuscript to PLOS ONE. After careful consideration, we feel that it has merit but does not fully meet PLOS ONE’s publication criteria as it currently stands. Therefore, we invite you to submit a revised version of the manuscript that addresses the points raised during the review process.

Please address reviewer and editor comments below. Regarding Reviewer #1's first comment, please cite and discuss Kirby and Vistnes(2016). 

We look forward to receiving your revised manuscript.

Kind regards,

Fernando A. Wilson, PhD

Academic Editor

PLOS ONE

Journal Requirements:

Additional Editor Comments (if provided):

- Please confirm that references conform to PLOS ONE requirements. For example, journal abbreviations should not include periods, i.e., use "N Engl J Med" instead of "N. Engl. J. Med". For further information, refer to: https://journals.plos.org/plosone/s/submission-guidelines#loc-references

Reviewers' comments:

Reviewer's Responses to Questions

**Comments to the Author**

1. Is the manuscript technically sound, and do the data support the conclusions?

Reviewer #1: Yes

Reviewer #2: Yes

2. Has the statistical analysis been performed appropriately and rigorously? 

Reviewer #1: Yes

Reviewer #2: Yes

3. Have the authors made all data underlying the findings in their manuscript fully available?

Reviewer #1: Yes

Reviewer #2: Yes

4. Is the manuscript presented in an intelligible fashion and written in standard English?

Reviewer #1: Yes

Reviewer #2: Yes

5. Review Comments to the Author

Reviewer #1: Thank you for the opportunity to review this paper. Using data from the Medical Expenditure Panel Survey (MEPS), the authors investigate the association between prior ‘insurance stability’ and change in use of emergency and office visits after the introduction of Medicaid expansion and Marketplace. The policy significance of this research question is obvious. The authors have used an appropriate data set and appropriate methods for their analysis.

However, the major issue with this paper is whether it makes sufficiently new contribution to the literature. The authors claim that “However, the relationship between how long someone went uninsured before gaining coverage, what we refer to as insurance instability, and subsequent changes in health care use after gaining coverage has not yet been studied” (lines 64-67). Unfortunately it’s not quite true. Kirby and Vistnes (2016) answered a very similar research question using the same data set and similar methods. The following is the abstract of that paper, from Health Affairs (https://www.healthaffairs.org/doi/10.1377/hlthaff.2016.0716

“Newly available longitudinal survey data show that people who lacked health insurance in 2013 and gained coverage through Medicaid or the Marketplaces in 2014 were far more likely to obtain a usual source of care and receive preventive care services than their counterparts who remained uninsured in 2014.”

Since this paper was not in the citations, I assume that the authors are not aware of this research. The only thing that seems to be different in this paper from the Health Affairs paper is that the authors are using different measures of health care use, though preventive service visits are likely a subset of office visits.

The authors therefore need to do a more careful review of the literature - incorporate the above-mentioned paper and make sure that they haven’t missed any other relevant study. They also need to convince the readers that their research in not just a minor extension of existing research.

Minor issues

1. The authors should exclude the 18-year-olds from the sample. The 18-year-olds have public coverage options from Medicaid or Children’s Health Insurance Program (CHIP) as children that are more generous than those for adults (even in the expansion states).

2. The authors made a good choice of excluding those with Medicare from the sample. Additionally, they should also exclude any one with Supplemental Security Income (SSI) as they are eligible for Medicaid as disabled, and the Medicaid program for the disabled have not changed as a result of the ACA.

Reviewer #2: PONE-D-20-09519 - Insurance instability and use of emergency and office-based care after gaining coverage: an observational cohort study

This paper uses linkage between the National Health Interview Survey (NHIS) and the Medical Expenditure Panel Survey (MEPS) to estimate the association between spells of unemployment until 2014 on changes in office-based and ED care in 2014 following insurance expansions from the ACA. The authors do a great job linking these and other datasets to adequately specify their models, and their four category outcome is sufficient to capture changes in treatment setting. Only a few minor points follow.

It seems that limited sample size may be the culprit for failing to precisely estimate some effects, particularly the substitution among adults gaining public insurance from long-term uninsurance. The authors state that they use separate regressions for short and long-term uninsured with the same reference group. The authors could consider pooling these treatments and allowing the other controls to be restricted or possibly interact with duration of uninsurance with certain covariates. This may lend some power, although it may not change the findings appreciably while adding some unneeded complexity.

The public vs. private analyses suffer from having what you can think of as non-compliers to treatment in each sample. Adults with income above 138% FPL cannot take-up public coverage (virtually entirely Medicaid in this case). The authors omit > 400% FPL, but should try also omitting > 138% FPL, sample size permitting. The authors could go further by subsetting to states that expanded Medicaid in 2014 if they are subsetting down to adults who had an opportunity to enroll in Medicaid. Likewise, the private could omit lower income more likely to take-up Medicaid.

As a minor point, the authors could include as a supplemental table the descriptives for all the variables used in the model, or at least those from the AHRF described in the paper. Also, table notes should include regression sample sizes where applicable.

6. PLOS authors have the option to publish the peer review history of their article (what does this mean?). If published, this will include your full peer review and any attached files.

Reviewer #1: No

Reviewer #2: No

---

## [Author Response · Author response to Decision Letter 0]

23 Jul 2020

We have uploaded a revised cover letter and point-by-point response to reviewers with this re-submission.

---

## [Editor Report · Decision Letter 1]

11 Aug 2020

Insurance instability and use of emergency and office-based care after gaining coverage: an observational cohort study

PONE-D-20-09519R1

Dear Dr. Shafer,

We’re pleased to inform you that your manuscript has been judged scientifically suitable for publication and will be formally accepted for publication once it meets all outstanding technical requirements.

Kind regards,

Fernando A. Wilson, PhD

Academic Editor

PLOS ONE
---

## [Editor Report · Acceptance letter]

17 Aug 2020

PONE-D-20-09519R1 

Insurance instability and use of emergency and office-based care after gaining coverage: an observational cohort study 

Dear Dr. Shafer:

I'm pleased to inform you that your manuscript has been deemed suitable for publication in PLOS ONE. Congratulations! Your manuscript is now with our production department. 

Kind regards, 

on behalf of

Dr. Fernando A. Wilson 

Academic Editor

PLOS ONE